# One Earth-One Health (OE-OH): Antibacterial Effects of Plant Flavonoids in Combination with Clinical Antibiotics with Various Mechanisms

**DOI:** 10.3390/antibiotics14010008

**Published:** 2024-12-25

**Authors:** Ganjun Yuan, Fengxian Lian, Yu Yan, Yu Wang, Li Zhang, Jianping Zhu, Aiman Fatima, Yuxing Qian

**Affiliations:** 1Biotechnological Engineering Center for Pharmaceutical Research and Development, Jiangxi Agricultural University, Nanchang 330045, China; 2Laboratory of Natural Medicine and Microbiological Drug, College of Bioscience and Bioengineering, Jiangxi Agricultural University, Nanchang 330045, China

**Keywords:** combination, flavonoid, antibiotics, mechanism, synergy, antimicrobial resistance, antibiotic resistance, one earth, one health, OE-OH

## Abstract

**Background/Objectives:** Antimicrobial resistance (AMR) poses a significant threat to human health, and combination therapy has proven effective in combating it. It has been reported that some plant flavonoids can enhance the antibacterial effects of antibiotics and even reverse AMR. This study systematically evaluated the synergistic effects of plant flavonoids and clinical antibiotics. **Methods**: The antibacterial activities of thirty-seven plant flavonoids and nine clinical antibiotics with various antimicrobial mechanisms were determined using the microbroth dilution method. Subsequently, the combined effects of twelve plant flavonoids presenting definite MICs against *Staphylococcus aureus* with these nine antibiotics were determined using the checkerboard test, together with those of thirty-two plant flavonoids presenting definite MICs against *Escherichia coli* with these nine antibiotics. **Results**: Plant flavonoids in combination with antibiotics present extensive synergistic effects, and 25% of combinations exhibited synergy against *S. aureus* and 50% against *E. coli*, particularly with antibiotics affecting cell membranes or ribosomes. **Conclusions**: The findings align with the drug selection principle of synergistic combinations and suggest that plant flavonoids could extensively enhance antibiotic efficacy. Considering that various metabolites from microorganisms, plants, and animals on the Earth would definitely impact the evolution of AMR, together with the rules, key factors, and important principles of drug combination for preventing AMR, we further propose the “One Earth-One Health (OE-OH)” concept, emphasizing ecosystem interactions in AMR prevention.

## 1. Introduction

Antimicrobial resistance (AMR) is a global crisis exacerbated by the overuse of antibiotics and a slowdown in new antibiotic discoveries [1,2,3], simultaneously the COVID-19 pandemic has further accelerated this global problem [3]. AMR and its evolution are very complex, involving many aspects [4,5]. For drugs, many strategies have been put forward to fight or delay resistance [6,7,8,9]. This indicates that rational combination therapies can not only enhance the clinical efficacy of antibacterial agents [10], but also make full use of clinical antibacterial resources to reduce the cost and gain enough time to prevent AMR and delay its evolution [7,10,11,12]. Simultaneously, many natural products from plants have been reported to possess antimicrobial properties and are generally considered as safe [13,14]. Therefore, combination therapy, particularly involving plant natural products, offers a promising strategy to restore antibiotic efficacy and slow down AMR progression.

As an important class of plant secondary metabolites, many plant flavonoids show different degrees of inhibitory activity against pathogenic bacteria, and some of them even have antibacterial activities comparable to clinical antibiotics [15,16,17]. The research on plant flavonoids in combination with clinical antibiotics has shown that some of them can not only remarkably enhance the antibacterial effects of clinical antibiotics, but can also reverse resistance or even enhance the susceptibility of pathogenic bacteria to clinical antibiotics [15,16,17]. Therefore, it is important to discover possible rules on the combination of plant flavonoids with antibiotics to quickly obtain synergistic combinations.

Simultaneously, it is generally believed that a combination of antibacterial agents with different mechanisms would present a higher probability of synergistic effects. However, the results evaluated by us indicate that most of them show non-synergistic antibacterial effects, and that antimicrobial agents targeting the same macromolecular biosynthesis pathway with different sites have a great potency to discover synergistic combinations [18]. Namely, the combination of antibacterial agents acting on different metabolic sites of the same biomacromolecule metabolic pathway would present a higher probability of synergistic effects. This result was immediately proven by Brochado et al. from the European Molecular Biology Laboratory in Germany [12] as well as by subsequent experiments on natural products [19]. Considering that only two compounds, α-mangostin and carnosic acid, were selected in the latter experiment, it is also essential to conduct more extensive validation to rapidly obtain synergistic combinations of plants with antibiotics.

Based on the two above-mentioned, here the synergistic effects of flavonoid–antibiotic combinations against pathogenic bacteria were systematically evaluated using 37 plant flavonoids with different antibacterial potentials and 9 clinical antibiotics with different antibacterial mechanisms, and *Staphylococcus aureus* and *Escherichia coli* were used as the representatives of Gram-positive and Gram-negative bacteria, respectively. Furthermore, combined with various laws and conclusions of combination therapy preventing AMR discovered by us [20], this research also underscores the role of natural ecosystems, leading to the “One Earth-One Health (OE-OH)” concept, a novel approach to AMR prevention proposed at the 6th International Caparica Conference in Antibiotic Resistance 2024 (IC^2^AR 2024) [21]. Now, the research is presented as follows.

## 2. Results

### 2.1. Minimum Inhibitory Concentration (MIC)

The MICs of nine clinical antibiotics against pathogenic bacterial *S. aureus* ATCC 25923 and *E. coli* ATCC 25922 are listed in Table 1. These antibiotics involve various antibacterial mechanisms including inhibition to the synthesis of cell wall or protein and the damage to cell membranes along with the alteration in membrane permeability. These have different activities against *S. aureus* and *E. coli*, with MICs ranging from 0.25 to 32 μg/mL and 1 to 1024 μg/mL, respectively.

Additionally, the MICs, expressed as the molar concentration (μM) of 37 plant flavonoids against *S. aureus* ATCC 25923 and *E. coli* ATCC 25922, were reported in our previous work [27], and here, the raw data of their MICs (μg/mL) were reorganized and are shown in Table 2. These plant flavonoids include various structural subtypes such as dihydroflavones, flavones, flavonols, chalcones, isoflavones, and xanthones. As seen in Table 2, they presented different antibacterial activity against *S. aureus* ATCC 25923, with MICs ranging from 2 to 4096 μg/mL or more than 2048 μg/mL, and a few of them showed antibacterial activities comparable to clinical antibiotics such as sophoraflavanone G and α-mangostin. However, all plant flavonoids in Table 2 showed weak inhibitory activities against *E. coli* ATCC 25922, and their MICs ranged from 512 to more than 2048 μg/mL.

### 2.2. Antibacterial Effects of Plant Flavonoids in Combination with Clinical Antibiotics to S. aureus

There were 12 plant flavonoids with definite MIC values against *S. aureus* ATCC 25923, as can be seen in Table 2: quercetin, anhydroicaritin, isovitexin, isoliquiritigenin, licoflavone C, rutin, naringenin, puerarin, glabridin, licochalcone A, sophoraflavanone G and α-mangostin. The antibacterial effects of these plant flavonoids in combination with nine clinical antibiotics (Table 1) were determined on 96-well plates, and the results are shown in Figure 1. Among these 108 combinations against *S. aureus*, 27 presented a synergistic effect, which was equal to 25% of all combinations.

In Figure 1, all of the tested plant flavonoids showed synergistic effects against *S. aureus* ATCC 25923 when combined with gentamicin sulfate or streptomycin sulfate, except for puerarin in combination with streptomycin sulfate. Simultaneously, a few of these plant flavonoids showed synergistic effects against *S. aureus* ATCC 25923 when combined with antibiotics that affect the cell membrane such as colistin sulfate or bacitracin. However, all of the tested plant flavonoids showed indifferent effects when combined with antibiotics that inhibit the biosynthesis of the bacterial cell wall or protein such as vancomycin hydrochloride, ampicillin, roxithromycin, doxycycline, and linezolid. As the antibacterial mechanism of gentamicin sulfate, streptomycin sulfate, colistin sulfate, and bacitracin involves the impact on the cell membrane, the above combinational effect of 12 plant flavonoids and 9 antibiotics is consistent with the selection rule of antibacterial agents for synergistic combinations [18].

### 2.3. Antibacterial Effects of Plant Flavonoids in Combination with Clinical Antibiotics to E. coli

As shown in Table 2, there were a total of 32 plant flavonoids with definite MIC values against *E. coli* ATCC 25922. The antibacterial effects of these plant flavonoids in combination with nine clinical antibiotics (Table 1) were also determined, and the results are shown in Figure 2. Among these 288 combinations against *E. coli*, 141 presented a synergistic effect, which was equal to 49.0% of all combinations. Namely, approximately half of these combinations presented a synergistic effect.

Unlike the combinational effects described in Section 2.2, here, these 32 plant flavonoids including most of the 12 flavonoids with definite MIC values against *S. aureus* ATCC 25923, in combination with antibiotics, showed extensive synergistic effects against *E. coli* ATCC 25922, as can be seen in Figure 2. Notably, all of the plant flavonoids showing a relatively stronger activity against *E. coli* ATCC 25922, such as glabridin, sophoraflavanone G, and α-mangostin, exhibited synergistic effects when combined with the antibiotics listed in Table 1. Additionally, isoliquiritigenin and licochalcone A also presented synergistic antibacterial effects with most of the tested antibiotics. It is worth noting that antibiotics clinically used for treating Gram-positive bacterial infections, such as vancomycin hydrochloride, linezolid, and bacitracin, showed synergistic effects against *E. coli* ATCC 25922 when combined with all of the tested plant flavonoids, although these antibiotics have weak activity against *E. coli*. Moreover, streptomycin sulfate showed synergistic effects against *E. coli* when combined with many of the tested plant flavonoids, which was similar to those cases against *S. aureus*. However, gentamicin sulfate exhibited synergistic effects against *E. coli* only when combined with glabridin, sophoraflavanone G, α-mangostin, isoliquiritigenin, licochalcone A, hispidulin, and naringin. The different effect of plant flavonoids in combination with antibiotics against *S. aureus* and *E. coli* may have been due to their different antibacterial mechanism [27].

### 2.4. Antibacterial Effects of Plant Flavonoids in Combination with Levofloxacin to E. coli

Considering that DNA gyrase is an important target for plant flavonoids against Gram-negative bacteria [27], plant flavonoids that presented extensive synergistic effects when combined with the tested antibiotics against *E. coli* including glabridin, sophoraflavanone G, α-mangostin, isoliquiritigenin, and licochalcone A are likely to exhibit non-synergistic effects in combination with quinolone antibacterial agents acting on DNA gyrase, according to the selection rule of antibacterial agents for synergistic combinations [18]. Therefore, the antibacterial effects of these five plant flavonoids in combination with levofloxacin against *E. coli* ATCC 25922 were further determined using the same methods described in Section 4.3 and Section 4.4. The results showed that all of the combinations exhibited indifferent effects, with FICI values ranging from 0.625 to 2.125. Conversely, this once again supports the rationality of the selection rule of antibacterial agents for synergistic combinations, namely that the probability of discovering synergistic combinations is higher from antibacterial agents that act on different metabolic sites of the same macromolecular metabolite pathway.

## 3. Discussion

Along with the increasing severity of AMR, the application advantages of natural products, particularly plant flavonoids, in resisting AMR have attracted much attention [15,16,17]. However, there are three aspects worth clarifying before being discussed. (1) Due to differences in the testing environment, conditions, methods, and specific operations, there are significant differences in the results reported from different literature sources for some compounds. Using the microbroth dilution method, the MIC value generally results from a series of concentrations with half dilution, and the actual one may not be exactly at the set concentration. For example, the series of concentrations may include 10, 5, and 2.5 μg/mL (or 12.5, 6.25, and 3.13 μg/mL), but if the observed MIC value is 5 μg/mL (or 6.25 μg/mL), the actual value could be 4 or 3 μg/mL (or 5 or 4 μg/mL). Therefore, an error of 1/2 to 2 × MIC would be introduced. Considering the differences in various laboratories, testing conditions and methods, and specific operations, even greater errors may result. Therefore, the ratio of MIC values reported for a compound against the same bacterial strain should be considered reasonable within the range of 1/2 to 2 and even 1/4 to 4 [28,29] using the microbroth dilution method. (2) There are significant differences in the inhibitory activities and/or action mechanisms for the same compound against Gram-positive and Gram-negative bacteria, respectively, due to their different cell structures, especially the bacterial envelope. However, some studies in the literature did not strictly differentiate these when reviewing the antibacterial mechanisms of plant flavonoids [15,27,29], which can easily lead to some confusion in antibacterial mechanisms, and the erroneous transmission of research results. (3) The antibacterial mechanisms of a few plant flavonoids are limited to molecular levels including only theoretical calculations with molecular docking, lacking comprehensive cellular experiments and actual explorations at the cellular biochemical level [28,30].

Based on these aspects, the differences in the antibacterial properties, combination therapy, and synergistic mechanisms of plant flavonoids in combination with antibiotics against Gram-positive and Gram-negative bacteria were discussed, and subsequently a new concept was also proposed.

### 3.1. Antibacterial Properties and Combination Therapy of Plant Flavonoids Against Gram-Positive and Gram-Negative Bacteria

As previous works have reported [27,28], Table 2 indicates that overall, plant flavonoids have stronger activities against Gram-positive bacteria than Gram-negative species. Simultaneously, plant flavonoids with stronger hydrophilicity exhibit stronger activities against Gram-negative bacteria, while those with stronger lipophilicity demonstrate strong activity against Gram-positive species. This indicates that there is a higher probability of discovering natural products with strong activity against Gram-positive bacteria from plant flavonoids. Therefore, the development of new antibacterial agents against Gram-positive bacteria from plant flavonoids should be encouraged.

More importantly, the results show that it is relatively easy to discover synergistic combinations of plant flavonoids and clinical antibiotics. Furthermore, the synergistic probability of plant flavonoids in combination with antibiotics against Gram-negative bacteria is about twice against Gram-positive species. This is very fortunate since there is a more severe resistance to Gram-negative bacteria than Gram-positive species to clinical antibiotics [31]. Therefore, increased research on the combined use of plant flavonoids with clinical antibiotics for discovering more synergistic combinations against Gram-negative bacteria, delaying the evolution of Gram-negative ones, is encouraged.

### 3.2. Drug Selection and Synergistic Mechanisms of Plant Flavonoids in Combination with Antibiotics

As previously reported, the cell membrane is the primary action site of plant flavonoids against Gram-positive bacteria, while there are multiple mechanisms of plant flavonoids against Gram-negative bacteria [27]. Besides the cell membrane being an important action site, DNA gyrase is also another important target of plant flavonoids against Gram-positive bacteria [27]. According to the results in Section 2.2, plant flavonoids showed extensive synergistic effects when combined with antibiotics acting on bacterial ribosomes and affecting the cell membrane such as gentamicin sulfate and streptomycin sulfate [19,23,24]. Simultaneously, a few plant flavonoids also showed synergistic effects when combined with colistin sulfate and bacitracin [22], which can damage the cell membrane of Gram-positive bacteria. In contrast, all of the tested plant flavonoids showed indifferent effects when combined with other antibiotics that do not target the cell membrane of Gram-positive bacteria. Given that the cell membrane is the main site of action of plant flavonoids against Gram-positive bacteria, involving cell membrane damage [29], the above antibacterial effects of plant flavonoids in combination with clinical antibiotics against *S. aureus* not only match the selection rule of antibacterial agents for synergistic combinations [18], but further prove the rationality of this rule in turn.

According to the results in Section 2.2, thee plant flavonoids showed universal synergistic effects when combined with antibiotics against *E. coli*, especially when combined with antibiotics such as vancomycin hydrochloride, bacitracin, and linezolid, which mainly inhibit Gram-positive bacteria and have weak activity against *E. coli*, with all combinations showing synergistic effects. This may be interpreted as the weak ability of these three antibiotics to penetrate the cell membrane and the outer membrane of *E. coli* to reach the inner membrane and cytoplasm where they act, but when combined with plant flavonoids that have membrane damage effects [17,32,33], the concentration of these antibiotics reaching the inner membrane and cytoplasm increases, thus presenting a synergistic antibacterial effect. This is also supported by the synergistic effects that resulted from plant flavonoids such as glabridin, sophoraflavanone G, and α-mangostin in combination with all of the tested antibiotics. As these three plant flavonoids have stronger activity against both *S. aureus* and *E. coli*, they not only have a stronger damage effect on the cell membrane [17,34,35], but can also reach the target site at the inner membrane or nuclear region of *E. coli* at higher concentrations since they have a stronger inhibitory activity against *E. coli*. This also may be due to the fact (see Figure 1) that, overall, the stronger the inhibitory activity of plant flavonoids against Gram-positive bacteria, the greater the probability of synergistic effects when they are combined with antibiotics against Gram-negative species, as these plant flavonoids likely have a greater membrane damaging effect [28,29]. However, for antibiotics that have a stronger inhibitory activity against *E. coli* or whose inhibitory activity against *S. aureus* and *E. coli* is approximate, this impact of plant flavonoids enhancing the membrane permeability of these antibiotics might be diminished since they can penetrate the outer membrane, or the site of action is out of the inner membrane of *E. coli*. This may be a biological explanation for the antibacterial effects of these antibiotics combined with plant flavonoids being consistent with the selection rule of antibacterial agents for synergistic combinations. These antibiotics include streptomycin sulfate, gentamicin sulfate, bacitracin, doxycycline, and ampicillin, and all of them involve an effect on the cell membrane [19,22,23,24,25]. In contrast, roxithromycin, which is mainly active against Gram-positive bacteria, showed indifferent effects when combined with plant flavonoids against *E. coli*. This result also followed the selection rule of antibacterial agents for synergistic combinations, and might be due to the main mechanism of action of roxithromycin against *E. coli* not necessarily or entirely being on the ribosome.

Based on above analyses, for antibiotics that have strong activity against Gram-positive bacteria but weak activity against Gram-negative ones, plant flavonoids can enhance the concentration of these antibiotics reaching their targets by damaging the cell membrane, thus exhibiting a synergistic effect. Simultaneously, antibiotics, except for quinolone antibacterial agents, in combination with plant flavonoids with stronger activities against both Gram-positive and Gram-negative bacteria, will likely show extensive synergistic effects due to their various mechanisms against Gram-negative bacteria and stronger damage to the cell membrane. Moreover, the combined antibacterial effects of plant flavonoids with antibiotics follow the selection rule of antibacterial agents for synergistic combinations [18]. Therefore, this rule also provides a theoretical basis to quickly discover synergistic combinations of plant flavonoids and antibiotics.

### 3.3. Clinical Antimicrobial Agents in Combination with Plant Flavonoids to Prevent AMR

Combination therapy can enhance the antimicrobial effects of antimicrobial agents. In the past, people mainly focused on the aspect enhancing the antimicrobial effects of antimicrobial drugs through combination therapy, with less attention to the prevention effect against bacterial resistance. However, synergistic combination or enhancing the antimicrobial effects does not mean that it can prevent AMR, although synergistic combinations are beneficial for preventing bacterial resistance [18,20]. Regardless of whether the combinational effect is synergy, indifference, or antagonism, one drug can always narrow the mutation selection window of another drug by increasing its dosage, according to our previous work [18,20], thereby achieving a prevention effect on AMR, according to the mutation selection window theory [36]. Of course, the more synergistic the combined effect, the greater the potential to prevent AMR, and the easier it is to manipulate [18,20]. However, synergistic combinations are, after all, rare, and can only delay the spread of AMR. Moreover, the abuse of drug combinations not only cannot prevent AMR, but may also accelerate the evolution and spread of AMR [12,37]. Given that plant flavonoids have good safety, antimicrobial activities, and extensively synergistic effects when combined with antibiotics, it is worth encouraging the use of combinations of plant flavonoids and antibiotics to prevent AMR. Additionally, plant flavonoids widely distributed in the diet through vegetables, fruits, and other foods will inevitably affect the in vivo antibacterial effect of antibiotics and the resistance of pathogenic bacteria to antibiotics [38].

### 3.4. Concept of the One Earth-One Health (OE-OH) to Prevent AMR

As previously described, many plant flavonoids have antimicrobial activity and exhibit widely synergistic effects when combined with clinical antibiotics. It is worth noting that these flavonoids are widely distributed in various plants across different habitats on the Earth including a variety of vegetables and fruits that are part of the diets of people in many countries worldwide. Additionally, plants contain many other secondary metabolites such as terpenoids, quinones, alkaloids, and other phenolic substances, many of which also have antimicrobial activity and present synergistic effects when combined with clinical antibiotics [32,33,39,40,41,42] such as ursolic acid, carnosic acid, emodin, berberine, and so on. Therefore, all of these plant secondary metabolites with antimicrobial activity would not only affect the in vivo antimicrobial effects of clinical antibiotics and the susceptibility of pathogenic bacteria to clinical antibiotics if they are transmitted into the human body [32,38,40], but can also have a significant impact on the spread of resistant bacterial populations caused by the discharge of antibiotics into various environments, since they exist in various plants on the Earth and are distributed in various ecosystems involving these plants.

Similarly, various environmental microorganisms on the Earth including those in humans and animals such as the gut microbiota can also produce various secondary metabolites [39]. These metabolites are not only important sources of clinical antibiotics [13], but also have the ability to inhibit or kill pathogenic bacteria. The in vivo and in vitro combined effects of antibiotics, each other from environmental microorganisms, also indicate that they may affect the susceptibility of pathogenic bacteria to other clinical antibiotics, thereby affecting the evolution of AMR. Therefore, various secondary metabolites with antibacterial activities would not only affect the in vivo antimicrobial effects of clinical antibiotics and the susceptibility of pathogenic bacteria to clinical antibiotics if they are metabolites from microorganisms in the human body or are transmitted into the human body from various foods, but can also have a significant impact on the spread of resistant bacterial populations caused by the discharge of antibiotics into various environments since they are produced by various environmental microorganisms on the Earth and are distributed in various ecosystems involving these microorganisms. In addition, the microorganisms, plants, and animals on the Earth can degrade and/or utilize various clinical antibiotics emitted into the Earth’s environment, which can reduce the accumulation of clinical antibiotics in the environment, thereby reducing the risk in antibiotic resistance and its transmission around the environment. Thereout, the entire ecosystem of Earth can have a significant impact on the evolution of AMR, whose impact may be positive or negative.

It can be deduced that the complexity and sufficient buffering capacity of the Earth’s ecosystem determines its sufficient self-regulation ability in the evolution of AMR [43,44,45,46]. Therefore, a balance between humans and pathogenic microorganisms could be ensured as long as unremitting efforts are made to minimize the abuse of antibiotics and use them as reasonably as possible. If this is achieved, the prediction from the World Health Organization for AMR by 2050 will not become a reality [47]. Based on the above discussions, together with the rules, key factors, and important principles of drug combination for preventing AMR [20], we proposed the concept of One Earth-One Health (OE-OH) for preventing AMR at the 6th International Caparica Conference in Antibiotic Resistance 2024 (IC^2^AR 2024) held in Portugal on 10 September 2024 [21].

From the above discussion in Section 3.1, Section 3.2, Section 3.3 and Section 3.4, the development of combinational research of antibiotics with natural products or plant extracts should be encouraged, and their clinical application should be attempted to obtain effective natural product–antibiotic combinations and alleviate the shortage situation of effective antibiotic resources. More importantly, strengthening the management of antibiotic use and relevant aspects, based on the OE-OH concept, may be an effective strategy to delay the evolution of AMR.

## 4. Materials and Methods

### 4.1. Antimicrobial Agents and Plant Flavonoids

Ten antibacterial agents were used to evaluate the combinational effect. Gentamicin sulfate (USP grade, 590 U/mg) was purchased from Shanghai Yuanye Bio-Technology Co., Ltd. (Shanghai, China); ampicillin (96%) was purchased from Shanghai Acmec Biochemical Co., Ltd. (Shanghai, China); linezolid (99%), colistin sulfate (≥19,000 U/mg), bacteriocin (>60 U/mg), streptomycin sulfate (98%), and levofloxacin (98%) were purchased from Shanghai Macklin Biochemical Co., Ltd. (Shanghai, China); vancomycin hydrochloride (900 ug/mg) was purchased from Meryer (Shanghai, China) Biochemical Technology Co., Ltd. (Shanghai, China); roxithromycin (USP grade, >940 U/mg) and doxycycline (USP grade, 88~94%), analytical pure for 3-(4,5-dimethyl-2-thiazolyl)-2,5-diphenyl-2*H*-tetrazolium bromide (MTT), and dimethyl sulfoxide (DMSO) were purchased from Sangon Biotech (Shanghai) Co., Ltd. (Shanghai, China).

Thirty-seven plant flavonoids were used to evaluate the combinational effect, and their chemical structures and sources have already been reported in our previous work [27]. Sophoraflavanone G (>98%) was purchased from Shanghai TopScience Co., Ltd. (Shanghai, China); naringin (95%), neohesperidin (≥98%), and hesperidin (95%) were purchased from Shanghai Yuanye Bio-Technology Co., Ltd. (Shanghai, China); rutin (≥98%) and methyl-hesperidin (95%) were purchased from Shanghai Acmec Biochemical Co., Ltd. (Shanghai, China); eriodictyol (≥98%), eriocitrin (≥98%), rhoifolin (≥98%), and licoflavone C (≥98%) were purchased from Wuhan ChemFaces Biochemical Co., Ltd. (Wuhan, China); hesperetin (97%), puerarin (98%), baicalein (98%), diosmin (95%), apigenin (≥95%), diosmetin (98%), galangin (98%), icaritin (>98%), isoliquiritigenin (98%), formononetin (98%), and naringenin (97%) were purchased from Shanghai Macklin Biochemical Co., Ltd. (Shanghai, China); didymin (≥98%), 5-demethylnobiletin (≥98%), 4′,5,7-trimethoxyflavone (≥98%), vitexin (≥98%), and isovitexin (≥98%) were purchased from Sichuan Weikeqi Biological Technology Co., Ltd. (Sichuan, China); narirutin (98%), α-mangostin (>98.0%), licochalcone A (>98.0%), nobiletin (≥98.5%), orientin (99%), isoorientin (98%), tangeritin (≥98.5%), quercitrin (98%), and sinensetin (98%) were purchased from Chengdu Push Bio-technology Co., Ltd. (Chengdu, China); quercetin (97%) and glabridin (99.8%) were purchased from Meryer (Shanghai) Biochemical Technology Co., Ltd. (Shanghai, China).

All of the aforementioned compounds were stored at −20 °C. Prior to use, they were dissolved in a specific volume of dimethyl sulfoxide (DMSO) or sterilized water (only for hydrochloride and sulfate of antibiotics), and then diluted with fresh sterilized Mueller Hinton broth (MHB) to achieve stock solutions with a concentration of 2048, 4096, 8192, or 16,384 μg/mL. Following this, the stock solution was thoroughly mixed and further diluted to the desired concentrations with sterile MHB immediately. Additionally, the concentrations of DMSO in all test systems was maintained at less than 5.0%, while the blank controls contained 5.0% DMSO.

### 4.2. Media, Bacterial Strains and Growth Condition

Casein hydrolysate was purchased from Qingdao Hope Bio-Technology Co., Ltd. (Qingdao, China), and starch soluble, beef extract, and agar powder were sourced from Sangon Biotech (Shanghai) Co., Ltd. (Shanghai, China). These reagents were employed in the preparation of the culture media. Mueller Hinton agar (MHA) was formulated with 17.5 g/L of casein hydrolysate, 1.5 g/L of starch soluble, 3.0 g/L of beef extract, and 17.0 g/L of agar powder, all dissolved in purified water, with a pH value adjusted to 7.40 ± 0.20. The Mueller Hinton broth (MHB) was prepared without agar powder, following the same composition and protocol as MHA.

*E. coli* ATCC 25922 and *S. aureus* ATCC 25923 were obtained from the American Type Culture Collection in Manassas, VA, USA. These bacterial strains were preserved in MicrobankTM microbial storage systems, supplied by PRO-LAB diagnostics in Toronto, Canada, at a temperature of −20 °C. Prior to use, both *E. coli* and *S. aureus* were cultured onto MHA plates at 37 °C. Subsequently, isolated pure colonies from these plates were transferred into MHB and incubated at 37 °C for 24 h on a rotary shaker (160 rpm). An aliquot of the overnight culture was then diluted 1:100 into fresh MHB and incubated at 37 °C until it reached the exponential growth phase, ready for subsequent experimental procedures. MHB was used for the antimicrobial susceptibility tests. All TopPette Pipettors, both the 2~20 μL and 20~200 μL models, were purchased from DLAB Scientific Co., Ltd. (Beijing, China).

### 4.3. Susceptibility Test

The MICs of 37 plant flavonoids against both pathogenic bacteria were reported in our previous work [27], together with their MICs (μM) of unit converted, and the raw data of their MICs (μg/mL) are reported here. Similarly, all of the MICs of the antimicrobial against both pathogenic bacteria were determined according to the standard procedure described by the Clinical and Laboratory Standards Institute (CLSI) [48]. Briefly, the exponential phase culture was diluted with MHB to achieve a bacterial concentration of approximately 1.0 × 10^6^ CFU/mL, and then the MICs against *E. coli* ATCC 25922 and *S. aureus* ATCC 25923 were determined using the broth microdilution method on the 96-well plates (Shanghai Excell Biological Technology Co., Ltd., Shanghai, China) in triplicate [27]. Based on the preliminary MIC values of the compounds, the initial concentration of 1024, 2048, or 4096 μg/mL was respectively established for the corresponding compound. Following a 24 h incubation of the 96-well plate at 35 °C, 20 μL of the MTT solution (4.0 mg/mL) was added into each well, then the plate was thoroughly shaken, and allowed to stand for 30 min at room temperature. The MIC, defined as the lowest concentration of the compound that completely inhibits bacterial growth in the micro-wells, was determined by the absence of color change, indicating no bacterial growth, in contrast to the sufficient bacterial growth observed in the blank wells, as described in reference [28].

### 4.4. Checkerboard Assay

Depending on the MICs of the plant flavonoids with exact MIC values and nine antibacterial agents, a checkerboard assay was designed to determine their FICIs in combinations against two pathogenic bacteria, according to a previous method [18], and the tests were performed on 96-well plates. Briefly, dilutions from 8 to 1/16 MIC for the plant flavonoids and antibacterial agents in the horizontal or vertical direction were prepared in a separate 96-well plate by the twofold dilution method. Next, a 100 μL dilution with different concentrations for two compounds in a combination was correspondingly added into the designed wells on another plate to obtain different proportions with the concentrations from 4 to 1/32 MIC of each compound. Columns 11 and 12 only contained MHB with 5 × 10^5^ cfu/mL bacterial strains used as blank controls. When the microbial growth in the blank wells was good at 35 °C for 24 h, the MIC was defined as the lowest concentration of bacterial growth visibly inhibited in the micro-wells. If necessary, the MTT stain was used to clearly observe the results like in Section 4.3. The MICs of two compounds alone were respectively observed from row A and column 1, and the MICs of two compounds in combinations were obtained from wells B2 to H8.

The FICs were calculated as per the following formula:$$FICI=\frac{{MIC}_{comb}^{A}}{{MIC}_{alone}^{A}}+\frac{{MIC}_{comb}^{B}}{{MIC}_{alone}^{B}}$$

Here, A and B are two compounds in a drug combination, ${MIC}_{comb}^{A}$ and ${MIC}_{alone}^{A}$ are the MICs of A in a combination and alone, respectively, and ${MIC}_{comb}^{B}$ and ${MIC}_{alone}^{B}$ are the MICs of B in a combination and alone, respectively.

The combining effect was interpreted as follows: synergy, FICI ≤ 0.5; indifference, 0.5 < FICI ≤ 4.0; and antagonism, FICI > 4.0 [49].

## 5. Conclusions

Based on above results, analyses, and discussion, the following conclusions were made as follows. (1) Plant flavonoids in combination with antibiotics present extensive synergistic effects, and it is easier to discover synergistic combinations of plant flavonoids and clinical antibiotics against Gram-negative bacteria than Gram-positive ones. (2) The combined effects of plant flavonoids with antibiotics follow the selection rule of antibacterial agents for synergistic combinations, which is likely due to the main mechanism of plant flavonoids damaging the cell membrane of Gram-positive bacteria and its multiple mechanisms on Gram-negative bacteria including membrane damage and the inhibition to DNA gyrase. (3) The “One Earth-One Health (OE-OH)” concept, emphasizing the role of natural ecosystems in preventing AMR evolution, was proposed as a novel approach to AMR prevention. It points out that microorganisms, plants, and animals on the Earth and their various metabolites would definitely have an impact on the evolution of AMR, however, the ecosystem of the Earth also has enough buffering capacity and self-regulation ability in the fight between human and pathogenic microorganisms.

## Figures and Tables

**Figure 1 antibiotics-14-00008-f001:**
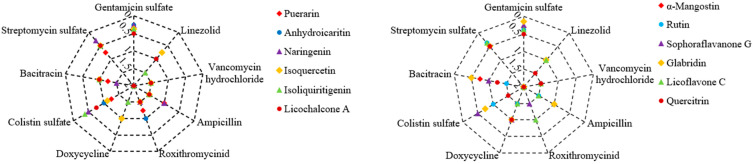
Radar chart showing the FICI (fractional inhibitory concentration index) values of 12 plant flavonoids in combination with 9 clinical antibiotics against *S. aureus*. Synergy is indicated by FICI ≤ 0.5.

**Figure 2 antibiotics-14-00008-f002:**
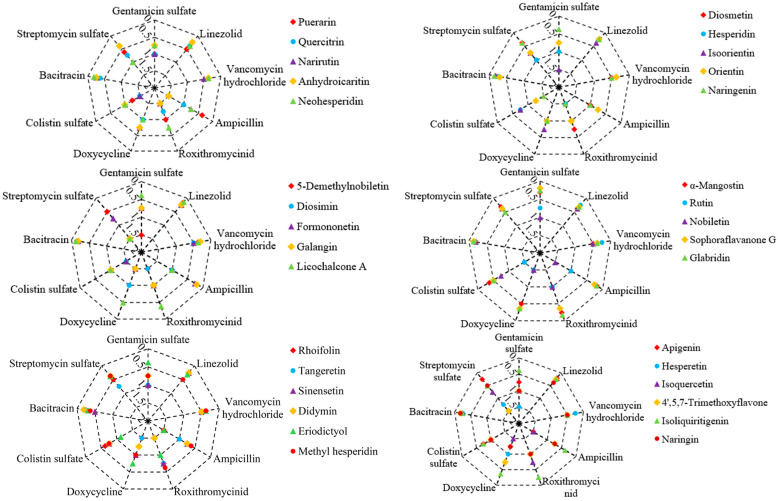
Radar chart showing the FICI values of 32 plant flavonoids in combination with 9 clinical antibiotics against *E. coli*. Synergy is indicated by FICI ≤ 0.5.

**Table 1 antibiotics-14-00008-t001:** The MICs of clinical antibiotics with various mechanisms of action against *S. aureus* and *E. coli*.

Antibacterial Agents	*S. aureus* ATCC 25923		*E. coli* ATCC 25922	
	The Subcellular Structure and Macromolecular Biosynthesis Pathway of Action	MIC (μg/mL)	The Subcellular Structure and Macromolecular Biosynthesis Pathway of Action	MIC (μg/mL)
Ampicillin	Binding to penicillin-binding proteins on the cell membrane, and inhibiting the biosynthesis of peptidoglycan of the cell wall	0.25	Same as *S. aureus*	2
Vancomycin hydrochloride	Targeting on the cell wall and inhibiting the biosynthesis of peptidoglycan	1	Difficult to reach the target site of action	512
Bacitracin	Targeting on the cell membrane and inhibiting the biosynthesis of peptidoglycan in cell wall [22]	32	Difficult to reach the target site of action	1024
Colistin Sulfate	Targeting on the cell membrane and increasing the permeability of the cell membrane	64	Targeting on the inner and outer membrane, increasing the permeability of the cell membrane, and affecting the stability of the cell membrane	1
Gentamicin sulfate	Targeting on the 30S subunit of bacterial ribosome and inhibiting the biosynthesis of protein, and affecting the permeability of cell membrane and the function of membrane proteins [19,23]	1	Same as *S. aureus*	8
Streptomycin sulfate	Same as gentamicin sulfate [24]	2	Same as *S. aureus*	1
Doxycycline	Targeting on the 30S subunit of bacterial ribosome and inhibiting the biosynthesis of protein [25]	0.5	Same as *S. aureus*	1
Roxithromycin	Targeting on the 50S subunit of bacterial ribosome and inhibiting the biosynthesis of protein	0.5	Unclear	64
Linezolid	Targeting on the 50S subunit of bacterial ribosome and inhibiting the biosynthesis of protein [26]	4	Difficult to reach the target site of action	256

**Table 2 antibiotics-14-00008-t002:** The MICs of 37 plant flavonoids against *S. aureus* and *E. coli* ^a^.

Plant Flavonoids ^b^	MICs (μg/mL)	
	*S. aureus* ATCC 25923	*E. coli* ATCC 25922
Quercetin	4096	>1024
Eriocitrin	>2048	>2048
Diosmetin, 5-demethylnobiletin, quercitrin, narirutin, orientin, isoorientin, rhoifolin, apigenin, hesperetin, sinensetin, didymin, eriodictyol, methylhesperidin, 4′,5,7-trimethoxyflavone	>2048	2048
Hesperidin, neohesperidin, tangeretin, naringin	>1024	2048
baicalein, vitexin	>1024	>1024
Formononetin, galangin, diosmin, nobiletin	>1024	1024
Anhydroicaritin, isovitexin, isoliquiritigenin	1024	2048
Licoflavone C	1024	>1024
Rutin	1024	1024
Naringenin	512	2048
Puerarin	256	2048
Glabridin	32	512
Licochalcone A	8	1024
Sophoraflavanone G	4	512
α-Mangostin	2	1024

^a^ The MICs, expressed as the molar concentration (μM), of 37 plant flavonoids against both pathogenic bacteria, were reported in our previous work [27]. ^b^ These plant flavonoids exist in many plants of various families such as Rutaceae, Verbenaceae, and Fabaceae.

## Data Availability

The MICs (μM) of 37 plant flavonoids against bacteria presented in Table 2 were reported in our previous work [27], and here, their raw data were reorganized and presented for the reference of the checkerboard experiment in Section 4.4. The structures of 37 plant flavonoids are also available from Figure 1 in our previous work [27].

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
