# Peer review of "One Earth-One Health (OE-OH): Antibacterial Effects of Plant Flavonoids in Combination with Clinical Antibiotics with Various Mechanisms"

_antibiotics, 2024, doi:10.3390/antibiotics14010008_

Round 1
Reviewer 1 Report
Comments and Suggestions for Authors
Review Introduction:
The manuscript titled "One Earth-One Health (OE-OH): Antibacterial effects of plant flavonoids in combination with clinical antibiotics with various mechanisms" addresses a critical topic in antimicrobial resistance (AMR), one of the most significant threats to global health. The authors aim to explore the synergistic effects of plant-derived flavonoids when combined with clinical antibiotics, particularly against Staphylococcus aureus and Escherichia coli, representatives of Gram-positive and Gram-negative bacteria, respectively.
The study is particularly valuable because it highlights the potential of natural plant compounds as enhancers of antibiotic efficacy. With AMR accelerating due to antibiotic misuse and limited discovery of new antibiotics, identifying natural synergistic combinations could pave the way for cost-effective and sustainable solutions. The authors also propose the "One Earth-One Health (OE-OH)" concept, emphasizing the role of ecosystems and natural metabolites in preventing AMR evolution, which adds a novel and holistic perspective to the fight against antibiotic resistance.
However, while the study is promising, the manuscript suffers from several issues that need attention. These include grammatical errors, redundancy, lack of clarity in certain sections, and formatting inconsistencies. The following review highlights these mistakes and provides detailed suggestions for improvement to enhance the manuscript's clarity, impact, and scientific rigor.
Identified Mistakes and Suggested Corrections:
1. Abstract
Issues:
The abstract is verbose and contains unnecessary redundancy.
The proposed "One Earth-One Health (OE-OH)" concept is mentioned abruptly.
Corrections:
Revised Abstract Example:
"Antimicrobial resistance (AMR) poses a major global health threat. This study investigates the synergistic effects of 37 plant flavonoids in combination with 9 clinical antibiotics against S. aureus and E. coli. Results showed that 25% of combinations exhibited synergy against S. aureus and 50% against E. coli, particularly for antibiotics affecting cell membranes or ribosomes. The findings align with drug synergy principles, suggesting that plant flavonoids could enhance antibiotic efficacy. We further propose the 'One Earth-One Health (OE-OH)' concept, emphasizing ecosystem interactions in AMR prevention."
2. Introduction
Issues:
The introduction lacks flow and clarity, particularly in explaining the significance of flavonoid-antibiotic combinations.
Redundant explanations of AMR and combination therapy weaken the introduction's focus.
Corrections:
Start with a strong statement on AMR's global impact.
Highlight the significance of plant-derived flavonoids and their potential to combat AMR.
Clearly state the study's objectives and introduce the OE-OH concept briefly without digressing.
Example:
"Antimicrobial resistance (AMR) is a global crisis exacerbated by the overuse of antibiotics and a slowdown in new antibiotic discoveries. Combination therapy, particularly involving natural compounds, offers a promising strategy to restore antibiotic efficacy and slow AMR progression. Plant-derived flavonoids, widely known for their antimicrobial properties, show potential to enhance clinical antibiotics' activity. This study systematically evaluates the synergistic effects of flavonoid-antibiotic combinations against Gram-positive and Gram-negative bacteria. The findings also underscore the role of natural ecosystems, leading to the 'One Earth-One Health (OE-OH)' concept, a novel approach to AMR prevention."
3. Figures and Tables
Issues:
Figures lack detailed captions.
Tables include repetitive content from the main text.
Corrections:
Add concise captions that summarize the key findings of each figure or table.
Remove redundant explanations in the text. For example:
Figure 1 Caption: "Synergistic effects (FICI values) of 12 flavonoids in combination with 9 antibiotics against S. aureus. Synergy is indicated by FICI ≤ 0.5."
4. Grammar and Sentence Structure
Issues:
Grammatical errors, such as: “AMR has been”
Here is a summary of the mistakes identified in the manuscript and suggestions for correction:
1. Abstract
Mistake: The abstract is overly long and contains redundant phrases, making it less concise.
Correction: Summarize findings more clearly and concisely. For example:
Original: "Our objective was to explore the combined effects of plant flavonoids with antibiotics."
Revised: "This study evaluated the synergistic effects of plant flavonoids and clinical antibiotics."
2. Grammar and Sentence Clarity
Examples:
Original: "Antimicrobial resistance (AMR) has been seriously threatening to human health..."
Correction: "Antimicrobial resistance (AMR) poses a significant threat to human health."
Original: "Combination therapy has been proved to be an effective strategy to fight the AMR."
Correction: "Combination therapy has proven effective in combating AMR."
3. Figures and Tables
Concern Figures and tables lack proper captions and clear legends.
Correction: Provide clear and descriptive captions for each figure/table, indicating what the reader should observe.
Example for Figure 1: "Radar chart showing FICI values of plant flavonoid combinations with clinical antibiotics against S. aureus. Synergy is indicated by FICI ≤ 0.5."
4. Technical Terms
Concern: Inconsistent usage of terms like "synergistic effect," "indifferent effect," and "antagonistic effect."
Correction: Ensure clear definitions of terms upfront and use them consistently throughout the manuscript.
5. Redundant Phrasing
Examples: Original: "The MICs (μg/mL) against bacteria presented in Table 2 are the raw ones of the MICs (μM) of 37 plant flavonoids."
Correction: "Table 2 presents MIC values (μg/mL) for 37 plant flavonoids."
Remove repetitive sentences across sections to improve flow.
6. Discussion Section
Concern: Some parts of the discussion repeat results without deeper interpretation.
Correction: Expand the discussion to:
Compare findings with previous studies.
Highlight the novelty of the study.
Suggest future research directions.
7. Conclusion
Concern: The conclusion introduces new ideas (e.g., "OE-OH concept") that are not fully elaborated in the results.
Correction: Focus on summarizing the main findings and their implications. Avoid introducing new concepts here unless they have been discussed earlier.
8. References
Concern: Some references are not cited in the text appropriately (e.g., "reported by us at MDPI").
Correction: Use consistent citation formatting (e.g., [Author et al., Year]) and avoid informal phrases like "reported by us."
Author Response
Dear Reviewer,
My co-authors and I are very grateful to you for your careful review, good comments, kind reminder, valuable suggestions, and great work and help. We have amended the manuscript according to the issues raised by you, and have pleasure to submit the revised version, together with the response to all points, for your consideration.
Please see the attachment.
Many thanks for your kind attention!
Yours sincerely,
Ganjun Yuan

Reviewer 2 Report
Comments and Suggestions for Authors
The author encapsulates in this article the significance of "exploring the synergistic effects of plant flavonoids in conjunction with antibiotics." The methodologies employed are articulated with clarity, and the results and discussion are generally presented in a coherent manner. Nevertheless, I would recommend the inclusion of representative plates from the MIC test prior to the paper's acceptance for publication.
Author Response
Dear Reviewer,
My co-authors and I are very grateful to you for your careful review, good comments, valuable suggestions, and great work. We have carefully checked the manuscript again, and have pleasure to submit the revised version, together with the response to your comments, for your consideration.
Please see the attachment.
Many thanks for your kind attention!
Yours sincerely,
Ganjun Yuan

Reviewer 3 Report
Comments and Suggestions for Authors
Dear Editor and the Authors,
The paper provided very detailed comparison of how common flavonoids and prescribed antibiotics show synergistic effect on gram (-) E.coli and gram (+) S.aureus. Even though, there are many papers describing how flavonoids contribute antibacterial activity of the common antibiotics, the current paper provides very detailed and extremely well presented results. Besides, the paper also described how MIC values can be decisive based on the starting concentration. The paper also has some philosophical aspect as one earth one health.. Even though, the paper can be accepted with the current form, if it is needed a revision, there can be one more sub section to show the key functional group differences of the flavonoids that possibly brought out the observed better synergy.
Kind Regards,
Author Response
Dear Reviewer,
My co-authors and I are very grateful to you for your careful review, good comments, kind reminder, valuable suggestions, and great work. We have amended the manuscript according to the issues raised by you, and have pleasure to submit the revised version, together with the response to your comments, for your consideration.
Please see the attachment.
Many thanks for your kind attention!
Yours sincerely,
Ganjun Yuan
